# Radical aryl migration enables diversity-oriented synthesis of structurally diverse medium/macro- or bridged-rings

Lei Li[1,*], Zhong-Liang Li[1,*], Fu-Li Wang[1], Zhen Guo[2], Yong-Feng Cheng[1], Na Wang[2], Xiao-Wu Dong[3], Chao Fang[4], Jingjiang Liu[4], Chunhui Hou[4], Bin Tan[1] & Xin-Yuan Liu[1]

Medium-sized and medium-bridged rings are attractive structural motifs in natural products and therapeutic agents. Due to the unfavourable entropic and/or enthalpic factors with these ring systems, their efficient construction remains a formidable challenge. To address this problem, we herein disclose a radical-based approach for diversity-oriented synthesis of various benzannulated carbon- and heteroatom-containing 8–11(14)-membered ketone libraries. This strategy involves 1,4- or 1,5-aryl migration triggered by radical azidation, trifluoromethylation, phosphonylation, sulfonylation, or perfluoroalkylation of unactivated alkenes followed by intramolecular ring expansion. Demonstration of this method as a highly flexible tool for the construction of 37 synthetically challenging medium-sized and macro-cyclic ring scaffolds including bridged rings with diverse functionalities and skeletons is highlighted. Some of these products showed potent inhibitory activity against the cancer cell or derivative of human embryonic kidney line in preliminary biological studies. The mechanism of this novel strategy is investigated by control experiments and DFT calculations.

[1] Department of Chemistry, South University of Science and Technology of China, Shenzhen 518055, China. [2] College of Materials Science & Engineering, Taiyuan University of Technology, Shanxi 030024, China. [3] College of Pharmaceutical Sciences, Zhejiang University, Hangzhou 310058, China. [4] Department of Biology, South University of Science and Technology of China, Shenzhen 518055, China. * These authors contributed equally to this work. Correspondence and requests for materials should be addressed to X.-Y.L. (email: liuxy3@sustc.edu.cn).

Collections and high-throughput screening of small-molecule libraries have become of paramount importance for discovering promising drug leads and biological probes. To meet this urgent need, the development of diversity-oriented synthesis approach for the rapid creation of complex molecules in the library has received increasing attention in both medicinal chemistry and chemical synthesis[1–3]. Although benzannulated medium- or medium-bridged rings constitute the basic skeletons of many important naturally occurring and biologically active molecules[4–9], they are really rare among the top 200 drugs[10]. The main reason is probably the difficulty in developing general strategies for the diversity-oriented synthesis of such library of structures[11–13]. The achievement of high efficiency remains a formidable challenge with conventional cyclization-based methods, mainly due to unfavourable enthalpic and entropic reasons. To address this, an alternative, more flexible strategy is the use of ring expansion reactions of bicyclic or polycyclic substrates to the desired medium-sized rings[14–19]. For example, Harrowven has reported a ring expansion approach involving radical *ipso*-substitution triggered by $sp^2$-radical intermediates generated from the dehalogenation of aryl or vinyl halides in the presence of toxic tin hydride reagents[16]. Recently, Tan and co-workers have developed an elegant biomimetic diversity-oriented synthesis of benzannulated medium-sized rings with rationally designed substrates via oxidative dearomatization followed by ring-expanding rearomatization (Fig. 1a)[18]. However, most reported methods rely heavily on the inherent nature of special substrates with preformed polycycles and/or highly strained small rings, which usually require tedious multistep preparation under harsh reaction conditions[14–19]. In addition, enantioselective ways to such chiral scaffolds have been rare because of the lack of efficient asymmetric approaches. Therefore, a general protocol for stereoselective and diversity-oriented synthesis of benzannulated medium- and related bridged-ring libraries, employing readily available starting materials and reagents under step/atom-economical and user-friendly synthetic conditions, is still highly desirable.

Unactivated alkenes are one of the most abundant feedstocks and thus represent excellent building blocks for chemical synthesis. The selective addition of diverse radicals to unactivated alkenes represents an efficient way for the functionalization of such alkenes[20,21]. In addition, great advances have been achieved in the field of aryl migration following radical *ipso* substitution at an aromatic ring for constructing new organic compounds[16,22–31]. Despite these advances, no radical-based protocol has been available for the conversion of unactivated alkenes into medium- to large-sized and related bridged ring libraries via aryl migration, especially for access to such chiral skeletons. To address these challenges, we rationally designed alkenyl alcohols **1** bearing cyclic benzyl alcohol groups[32,33], which are more readily accessible than those polycyclic substrates required in conventional methods[14–19] (Fig. 1b). It is expected that an inherently high-energy $sp^3$-carbon-centred alkyl radical **A** could be *in situ* generated from the addition of appropriate radicals to the unactivated alkenyl moiety of **1**. Driven by the formation of a lower energy neutral ketyl radical **C** (refs 32–34), intermediate **A** would undergo an intramolecular radical remote 1,4- or 1,5-aryl migration/ring expansion sequence to provide desired medium-ring products (Fig. 1c). Furthermore, we envisioned that the use of easily available optically pure alcohols would lead to enantioenriched medium-sized rings through a remote chirality transfer strategy. Hence, this strategy can overcome the unfavourable entropic and/or enthalpic factors typically encountered by conventional cyclization-based approaches[11–13]. Nonetheless, the realization of such cascade sequence reactions remains an underdeveloped process. Several challenges are

associated with the development of this reaction, such as (1) the unfavourable kinetic and/or thermodynamic factors to realize regioselective medium ring formation over other competitive 1,2-difunctionalizations of alkenes with oxygen-based nucleophiles[35–38], (2) the compatibility between unactivated alkenyl substrates and various radical precursors with different reactivity properties, (3) the identification of mild reaction conditions to achieve high degree of enantiocontrol through remote chirality transfer strategy.

With this strategy, we report herein a practical strategy for diversity-oriented synthesis of benzannulated 8-11(14)-membered cyclic ketones along with concurrent installation of various functional groups from readily available starting materials. This strategy was realized through a concerted remote 1,4- or 1,5-aryl migration/ring expansion sequence triggered by radical azidation, trifluoromethylation, phosphonylation, sulfonylation or perfluoroalkylation of unactivated alkenes (Fig. 1b, path 1). Given the prevalence of medium-bridged amines and relative chiral compounds in medicinal chemistry[39,40], we further demonstrate a distinct pathway for the convenient collection of useful medium-bridged amines with a high degree of skeletal complexity and functional diversity (Fig. 1b, path 2). In the context of a diversity-oriented synthesis of medium- and bridged-ring library, our strategy has displayed some exceptional advantages: (1) 1,4- and 1,5-aryl migration could be realized with the easily accessible substrates. (2) This method tolerates different opening-ring sizes of carbo- and heterocyclic alcohols from 5 to 11-membered ring, thus leading to a skeletally diverse medium- and large-ring collection. (3) A variety of radical sources, including azide, trifluoromethyl, phosphonyl, sulfonyl, perfluoroalkyl radical, are used as suitable radical precursors. (4) The functional groups in formed products serve as versatile handles for further diversification to afford diverse natural analogues in parallel. Therefore, this discovery will be a highly flexible tool for diverse collection of a variety of medium-sized rings and related bridged ring libraries with over 37 distinct scaffolds as well as the identification of some compounds with potent inhibitory activity against the cancer cell or derivative of human embryonic kidney lines. Noteworthy is that many formed benzannulated medium-sized or medium-bridged rings constitute the key synthetic intermediates and unnatural analogues of many important naturally occurring and biologically active molecules such as benzannulated cyclooctanes (antiarrhythmic, antidiarrheal, antiallergic and anti-inflammatory)[4], nonadride family (byssochlamic acid, heveadride, rubratoxin B, glaucanic and glauconic acid with antifungal, antimicrobial and antitumor activities)[5,6], isopavine family (neurological disorders inhibition)[7,8] and colchicine derivatives (anticancer activity)[9] (Fig. 2).

## Results

**Azidation-initiated medium/macro-sized ketones' synthesis.** Modular synthetic access to these model substrates **1** would be readily achieved in good yields by straightforward and efficient one-step transformation from inexpensive starting materials (see details in Supplementary Information). Since the development of methods for the incorporation of an azide moiety into organic compounds has enormous significance in chemistry, biology and materials sciences[41,42], we started our investigation to target the challenging nine-membered ketone formation along with the concurrent installation of an azide group based on the proposed pathway. As such, the reaction of substrate **1A** with iodine (III) reagent azidoidinane (**2a**), first reported by Zhdankin and co-workers[43], was selected for the optimization of reaction conditions (Supplementary Table 1). We were delighted to find that the desired nine-membered product **3A** was indeed observed

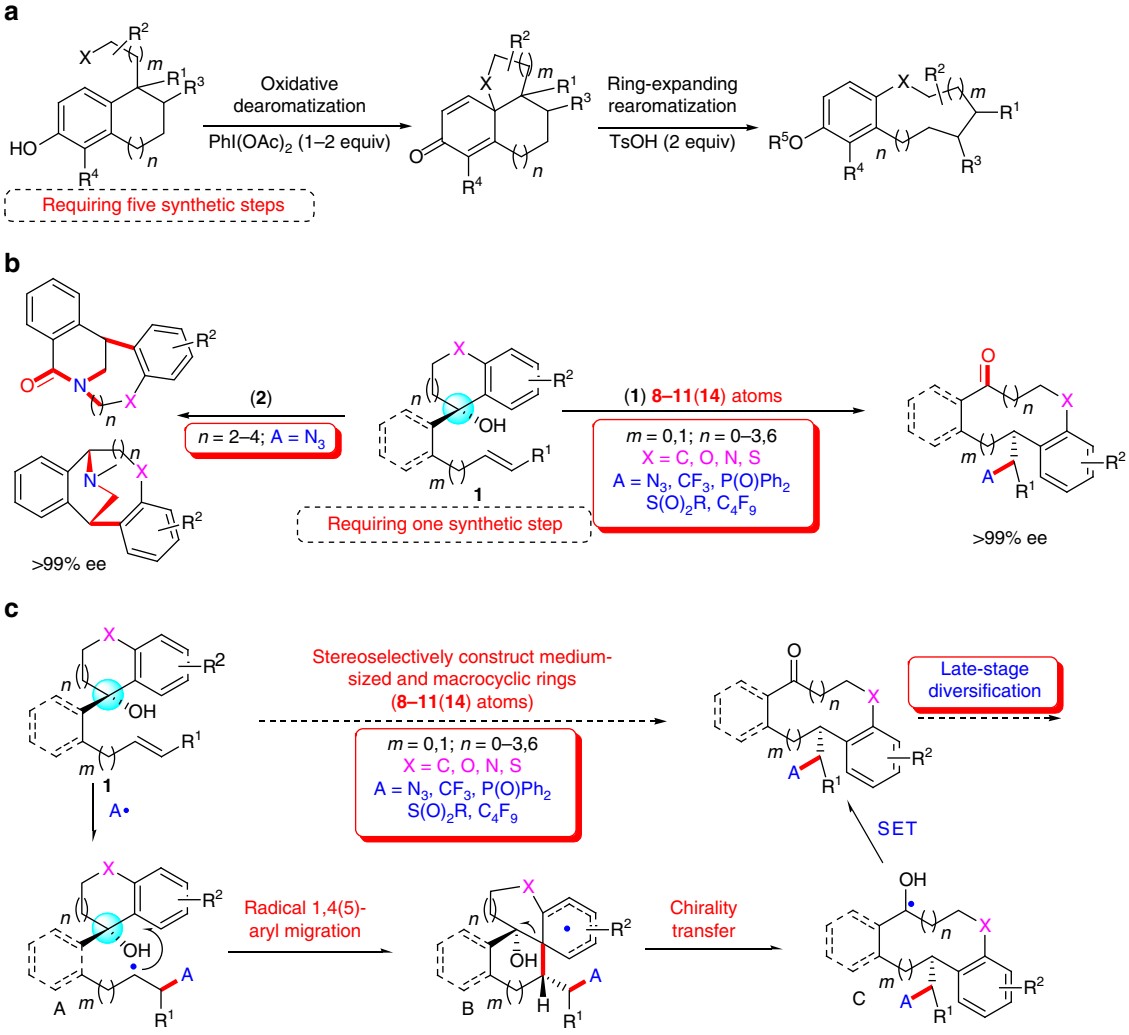

**Figure 1 | Radical strategy for asymmetric construction of medium-sized ketones and bridged amines.** (**a**) Previous work for construction of benzannulated medium-sized rings via ring expansion[18]. (**b**) Our radical-based strategy for construction of benzannulated medium- to large-sized ketones and bridged amines. (**c**) Proposed mechanism for the stereoselective construction of medium-sized rings.

in the presence of CuI (10 mol%), albeit with 47% yield, suggesting that the remote radical 1,4-aryl migration is much more favourable than other reaction pathways in the current catalytic system. Upon optimizing the reaction conditions through variation of the copper catalysts, catalyst loadings, solvents and the molar ratio of the reactants, we identified the following protocol as optimal: reaction of **1A** and **2a** with the molar ratio of 1.0:1.2 in the presence of CuCN (10 mol%) in EtOAc at 60 °C for 12 h, **3A** was obtained in 72% isolated yield (Table 1). With the optimized reaction conditions established, we first examined the substrate scope of alkenyl alcohols **1** featuring a six-membered ring and the results are summarized in Table 1. A range of substrates with 1,2,3,4-tetrahydronaphthalene moieties bearing mono-substituents, such as electron-withdrawing (-Br (**1A**), -F (**1B**)), electron-neutral (-H (**1C**)), and electron-donating (-OMe (**1D**)) groups, were found to be suitable to give the corresponding products **3A**-**3D** in 41–72% yields. Moreover, the reaction of **1E** with 2,5-dimethyl groups on the aryl ring proceeded smoothly to generate **3E** (49% yield). Remarkably, the products **3A**-**3E** are quite similar to the core structure of key intermediates towards the synthesis of glaucanic acid[6]. It is noteworthy that the alkenyl alcohols with heteroatom-tethering groups, such as sulfonamide- and sulfur-tethered substrates **1F**

and **1G**, were also well tolerated to produce heterocyclic ketones **3F** and **3G** in 76% and 68% yields, respectively. To increase the diversity of such a library, the more challenging benzannulated 10- and 11-membered ketone products **3H**-**3J** were also obtained in 74–82% yields from the corresponding substrates under the current reaction system. The high efficiency of the present protocol in preparing macrocyclic ring system was demonstrated in the isolation of 14-membered ketone **3K** in 65% yield from 11-membered alkenyl alcohol **1K**, which further extended the compound library. Notably, substrate **1L** containing another reactive olefin also afforded the desired 14-membered alkenyl ketone **3L** in 56% yield with the additional olefin being intact. Meanwhile, the alkenol **1M** containing an internal alkene in the presence of 1.5 equiv of 1,10-phenanthroline afforded the desired product **3M** as a 1.5:1 mixture of diastereomers in 55% yield.

**Construction of trifluoromethylated medium-sized ketones**. The increasing importance of trifluoromethyl organic molecules in agrochemicals, pharmaceuticals and molecular materials has spurred vigorous research efforts towards new versatile methodologies for the generation of the C-CF₃ bond[44,45]. Impressive

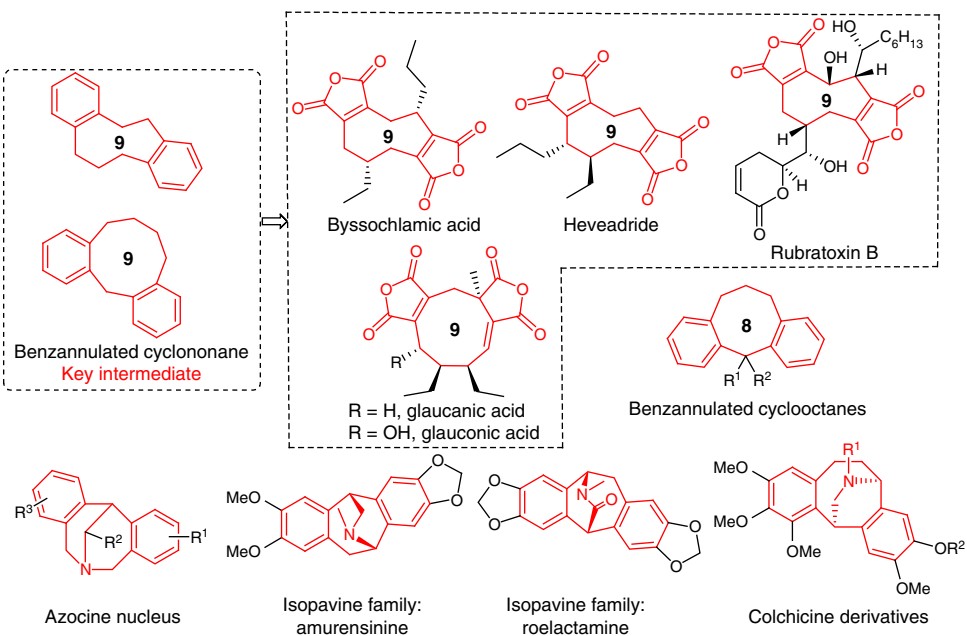

**Figure 2 | Representative natural products and biologically active molecules with benzannulated medium-sized rings and bridged amines.** Several core structures or their derivatives of natural products are synthesized in next section.

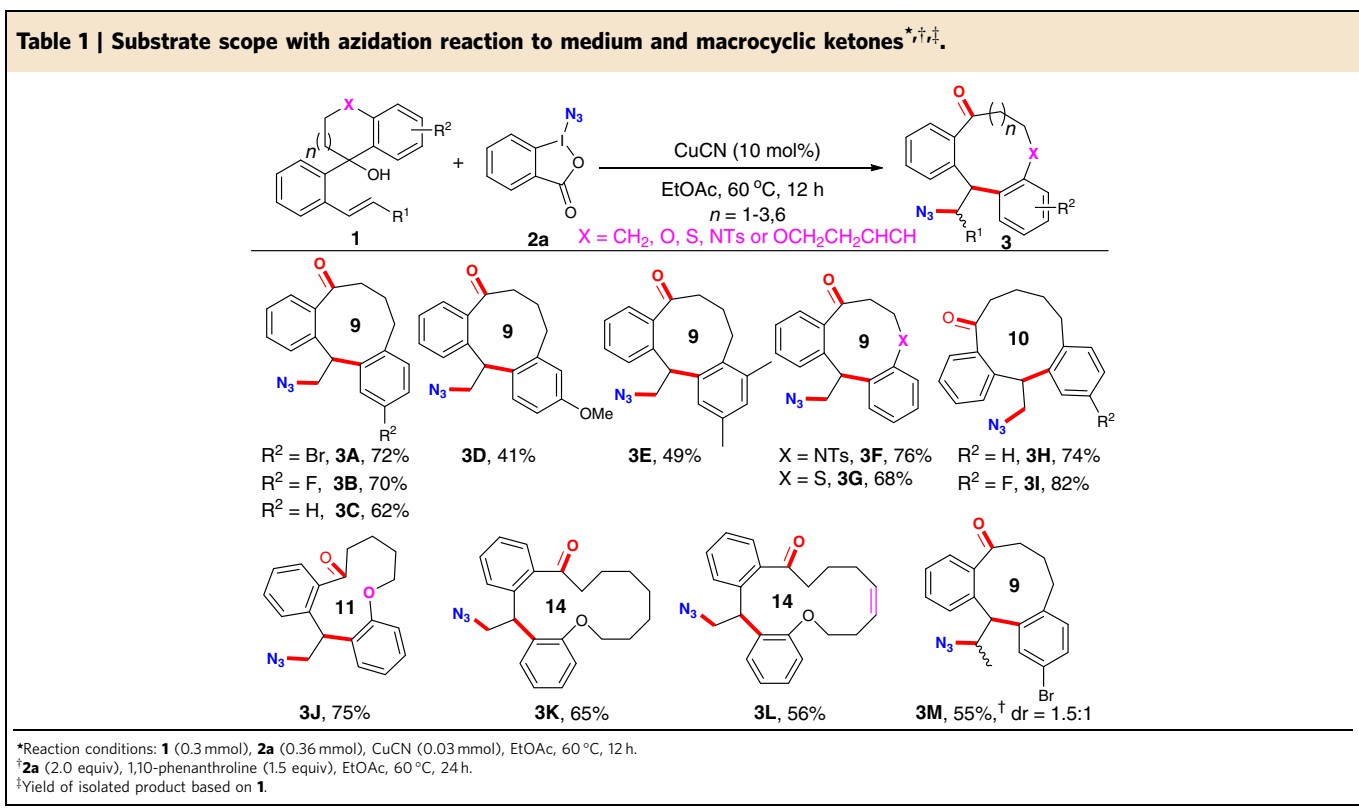

**Table 1 | Substrate scope with azidation reaction to medium and macrocyclic ketones\*,†,‡.**

*Reaction conditions: **1** (0.3 mmol), **2a** (0.36 mmol), CuCN (0.03 mmol), EtOAc, 60 °C, 12 h.
†**2a** (2.0 equiv), 1,10-phenanthroline (1.5 equiv), EtOAc, 60 °C, 24 h.
‡Yield of isolated product based on **1**.

advances have been achieved in developing methods for direct introduction of the CF$_3$ group into organic compounds by using radical trifluoromethylating reagents[35,36]. In this context, we focused our attention on realizing the incorporation of a CF$_3$ group with concurrent construction of benzannulated medium-sized rings (Table 2). To our delight, after some optimization efforts (Supplementary Table 2), we found that the reaction of **1A** with Togni's reagent[46] **2b** (2.0 equiv) proceeded smoothly in the presence of CuCN (10 mol%), giving the desired product **4A** in

64% yield with high chemoselectivity (Table 2a). A variety of alkenyl alcohols **1**, bearing either electron-withdrawing or electron-donating groups on the phenyl ring, reacted smoothly with **2b** to give the expected products **4B**-**4D** in 49–61% yields. Notably, the alkenyl alcohols with heteroatom-tethering groups (**1F** and **1G**) or seven-membered cycles (**1H** and **1I**) were also suitable substrates to deliver nine- and ten-membered hetero- or carbocyclic ketones **4F**-**4I** (35–75% yields). To verify the application of the current method in the synthesis of the core

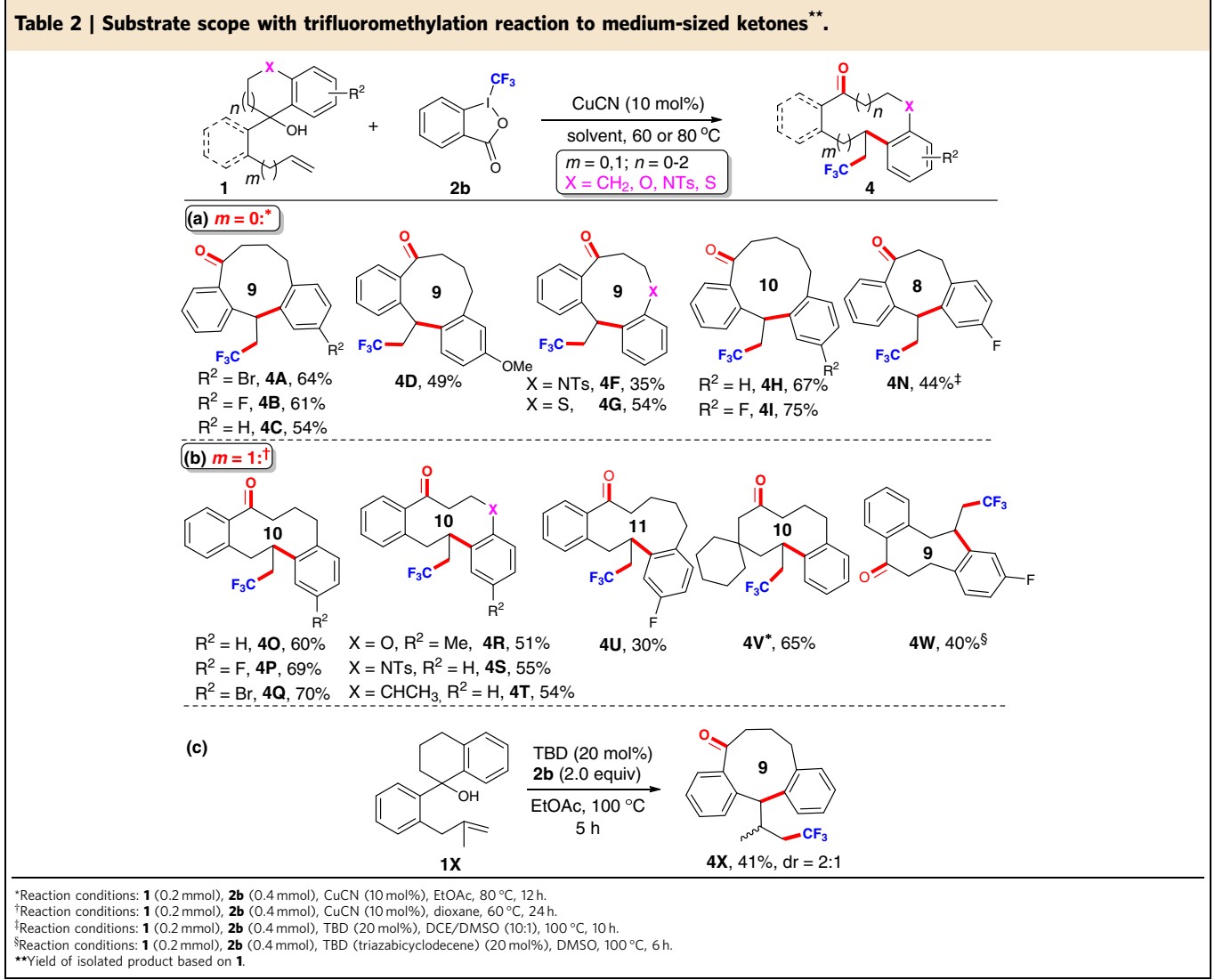

**Table 2 | Substrate scope with trifluoromethylation reaction to medium-sized ketones[**].**

*Reaction conditions: **1** (0.2 mmol), **2b** (0.4 mmol), CuCN (10 mol%), EtOAc, 80 °C, 12 h.
†Reaction conditions: **1** (0.2 mmol), **2b** (0.4 mmol), CuCN (10 mol%), dioxane, 60 °C, 24 h.
‡Reaction conditions: **1** (0.2 mmol), **2b** (0.4 mmol), TBD (20 mol%), DCE/DMSO (10:1), 100 °C, 10 h.
§Reaction conditions: **1** (0.2 mmol), **2b** (0.4 mmol), TBD (triazabicyclodecene) (20 mol%), DMSO, 100 °C, 6 h.
**Yield of isolated product based on **1**.

skeleton of the natural occurring benzannulated cyclooctanes[4] (Fig. 2), we turned our attention to target benzannulated eight-membered ketone **4N** and tested substrate **1N** with a five-membered ring under the standard conditions. To our disappointment, such substrate preferentially underwent competitive 1,2-oxytrifluoromethylation[37,38] to give product **4N′** in 65% yield with good diastereoselectivity (14:1 d.r.) (Supplementary Table 3). This observation clearly indicated that tuning the opening-ring size can have a profound influence on the control of the reaction outcome, probably owing to the presence of different favourable conformations and transition states during the reaction. To modulate the reactivity, we resorted to our recently developed organic base-catalysed radical trifluoromethylation of alkenes[33], and obtained the desired eight-membered product **4N** in 44% yield in the presence of triazabicyclodecene (TBD) (20 mol%) in a mixed solvent system of 1,2-dichloroethane (DCE)-dimethyl sulfoxide (DMSO) (10:1) (Supplementary Table 3).

To expand the synthetic utility of this methodology for access to other novel benzannulated medium-sized skeletons, we explored the use of allylbenzene derivatives as the substrates in place of styrene derivatives (Table 2b). After some experimentation (Supplementary Table 4), such a process for the formation of benzannulated ten-membered ketones could be successfully

realized directly from readily available alkenyl alcohols. For example, substrates **1** bearing various functional groups (**1O**-**1Q**) on the aryl ring, heteroatom-tethering groups (such as oxygen- and sulfonamide-tethered substrates **1R** and **1S**), and methyl-substituted 1,2,3,4-tetrahydronaphthalene group (**1T**), reacted efficiently to afford the corresponding products **4O**-**4T** in 51–70% yields. The structure of the product **4O** was also established by X-ray crystallographic analysis (Supplementary Fig. 1). Particularly noteworthy is that the 11-memerbed ketone **4U** could also be constructed with the current process, albeit in moderate yield with substrate **1U** bearing a seven-membered ring. It is noteworthy that the current method could be extended to the aliphatic substrate **1V**. Under the standard reaction conditions, product **4V** was obtained in 65% yield, indicating that the reaction was not significantly affected by switching the benzylic alcohol (**1O**) to an aliphatic substrate (**1V**). The resemblance of **4W** to the core structure of a key intermediate towards the synthesis of byssochlamic acid analogues[5] prompted us to test substrate **1W**. It was found that **4W** was generated smoothly as well, further elaborating the powerful utility of the current method in natural product synthesis. An interesting result was that the reaction of *gem*-disubstituted alkenyl alcohol **1X** generated the unexpected product **4X** in 41% yield as a 2:1 mixture of diastereomers, probably via a 1,2-hydrogen atom

**Figure 3 | Addition of diverse radicals to alkenes for the construction of medium-sized ketones.** (**a**) Radical alkene phosphonylation-initiated reaction. (**b**) Radical alkene sulfonylation-initiated reaction. (**c**) Radical alkene perfluoroalkylation-initiated reaction.

transfer of intermediate **A** followed by a 1,4-aryl migration process (Table 2c).

**Synthetic application towards diverse radicals**. The high efficiency of the above diversity-oriented synthesis approach encouraged us to expand the scope to other radical precursors. As expected, the reaction of substrate **1Q** with $Ph_2P(O)H$ (**5a**, 2.0 equiv) in the presence of $AgNO_3$ (0.5 equiv) delivered phosphonyl-containing ten-membered ketone **6** (Fig. 3a and Supplementary Table 5). Furthermore, the phosphonyl-containing ten-membered ketone **7** with a sulfonamide-tethering group was obtained in 65% yield. Moreover, dibenzylphosphine oxide **5b** also participated in this reaction to give product **8** in 66% yield, suggesting the feasibility of other types of phosphonyl radicals in this reaction. In addition to phosphonyl radical, *in situ* generated sulfonyl radical was also a suitable reaction partner: the reaction of substrate **1I** with 1.5 equiv of 3,5-bis(trifluoromethyl)benzenesulfonyl chloride **9** in the presence of CuI (10 mol%) and $AgNO_3$ (0.75 equiv) afforded the corresponding ketone **10** in 69% yield (Fig. 3b). The final success of $C_4F_9$ radical installation to generate **12** with perfluorobutanesulfonyl chloride (**11**, 1.5 equiv) via extrusion of sulfur dioxide under the otherwise identical reaction conditions was in great support of the wide application of the present methodology with different radicals in enriching the library of medium- and large-sized compounds (Fig. 3c).

**Chirality transfer strategy and mechanistic investigations**. On the basis of the concerted nature of the intramolecular radical aryl migration/ring expansion process as proposed in Fig. 1c, we expected that the stereochemical information of the tertiary alcohol would be completely transferred to the remote new-formed carbon chiral centre in a highly stereoselective way[47,48] to construct optically pure medium-sized ketone. However, this chirality transfer strategy might be difficult due to the presence of several highly reactive radical intermediates, as well as the requirement of discrimination between the two enantiotopic faces

in the intermediate **B**. To our delight, we found that the desired product (S)-**3A** was obtained in 70% yield with 99% ee via an almost complete chirality transfer strategy upon treatment of (R)-**1A** (99% ee) with **2a** under the otherwise identical conditions (Fig. 4a). Remarkably, almost complete chirality transfer could also be achieved for substrates (R)-**1A** and (S)-**1A** upon treatment with Togni's reagent **2b** under the standard conditions to give (S)-**4A** and (R)-**4A** with 99% ee (58% yield) and 98% ee (60% yield), respectively, thus indicating the present transformation is highly stereoselective (Fig. 4a). The absolute configurations of the substrate (R)-**1A** and the product (R)-**4A** were established by X-ray crystallographic analysis (Supplementary Fig. 1), and those of other materials and products were assigned by analogy (Fig. 4a). To further evaluate the practicality of this methodology, we expanded the substrate scope with the enantiomerically pure allylbenzene derivative **1Q\***. Treatment of substrate **1Q\*** with **2b** or **5b** under the otherwise identical conditions also underwent almost complete chirality transfer to give the corresponding enantioenriched benzannulated ten-membered trifluoromethyl- and phosphorus-containing ketones **4Q\*** and **8\*** in 64% and 70% yields with 96% and 95% ee, respectively (Fig. 4b,c).

To gain some more insights into the reaction mechanism, we tested some model reactions in the presence of radical scavengers such as 2,2,6,6-tetramethyl-1-piperidinyloxy (TEMPO) (for azidation, trifluoromethylation or phosphonylation reaction), 2,6-di-*tert*-butyl-4-methylphenol (BHT) (for azidation reaction) or 1,4-dinitrobenzene (for trifluoromethylation or phosphonylation reaction) under the standard conditions, and found that the reactions were significantly inhibited by these reagents (Supplementary Fig. 2), suggesting that a radical process is involved in these reactions. To probe the origin of the observed stereoselectivity, we further investigated the reaction mechanism computationally using M11 method. The generally assumed alkyl $sp^3$-carbon-centred radical species **1Q₁** was chosen as the starting point to locate two reaction pathways responsible for the stereoselective reactions. The calculated results revealed that this reaction occurs stepwise, involving the formation of the bicyclic

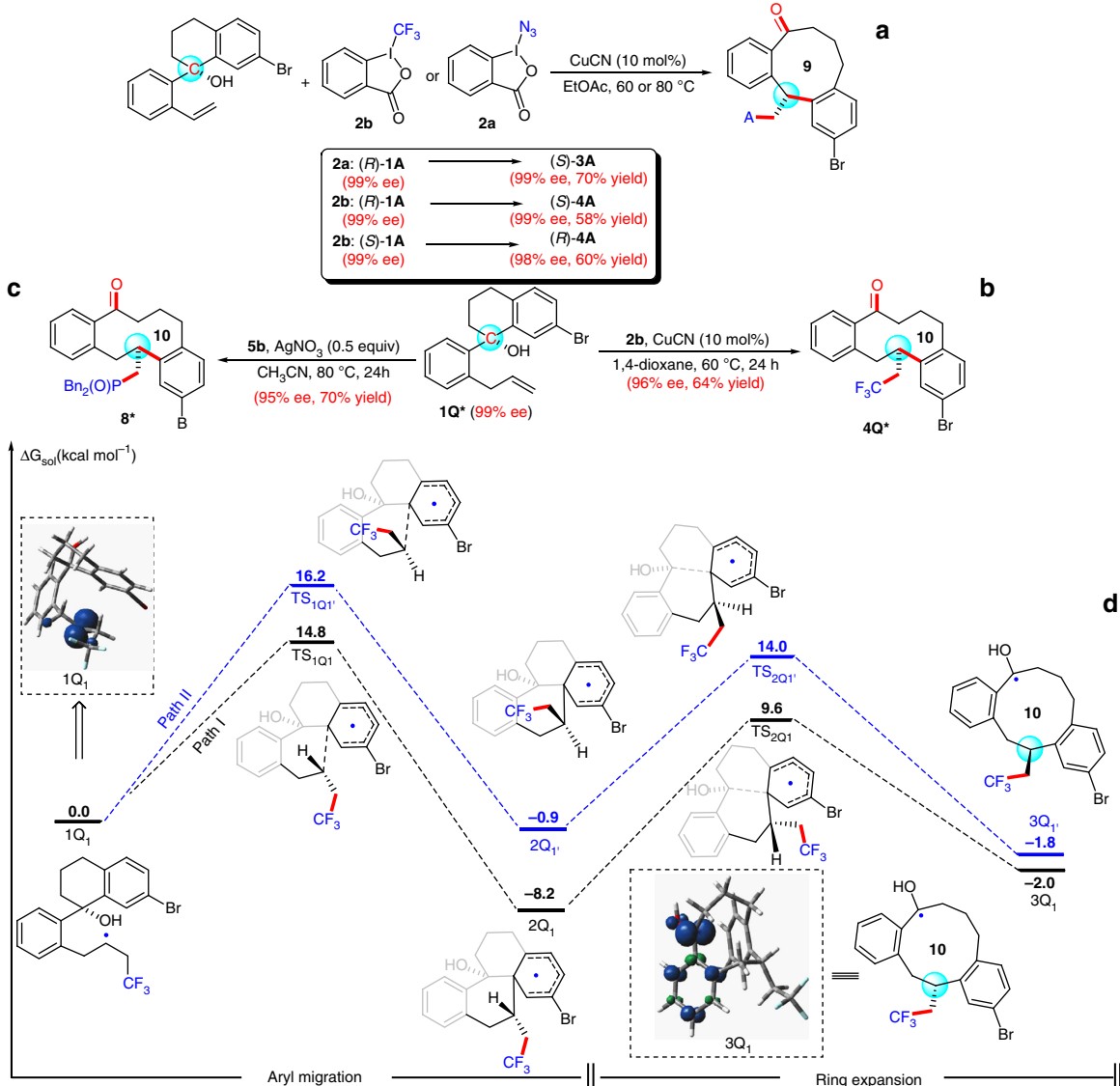

**Figure 4 | Reactions for enantioenriched medium-sized ketones through chirality transfer strategy and DFT calculation. (a)** Access to enantioenriched azido- and trifluoromethyl substituted medium-sized ketone from enantiopure **1A**. **(b,c)** Access to enantioenriched trifluoromethyl- and phosphonyl substituted medium-sized ketone from enantiomerically pure **1Q\***. **(d)** DFT calculation for mechanistic investigation. The calculated potential energy surfaces for aryl migration and ring expansion processes at M11/BS level of theory in 1,4-dioxane with the full optimization using smd model (BS refers to the used basis sets. For C, H, O, F atoms, the $6-31+G^{**}$ basis set was used and for Br atoms, the Aug-cc-PVTZ basis set was used).

rings $2Q_1/2Q_{1'}$ and the ring expansion to result in the formation of intermediates $3Q_1/3Q_{1'}$ (Fig. 4d). Some notable points from these calculations are as follows: (i) The addition of $sp^3$-carbon-centred radical species to the aryl group (from $1Q_1$ to $2Q_1/2Q_{1'}$) is exothermic. In addition, the reversed process ($2Q_1/2Q_{1'}$ to $1Q_1$) has higher reaction barriers than the subsequent ring expansion ($2Q_1/2Q_{1'}$ to $3Q_1/3Q_{1'}$). As such, reaction barriers of the aryl migration step could be used to mirror the observed stereoselectivity. (ii) The energy of transition state $TS_{1Q1}$, leading to the major ($S$)-product $3Q_1$, is lower than that of transition state $TS_{1Q1'}$ forming the minor ($R$)-product $3Q_{1'}$ by 1.4 kcal/mol when ($R$)-$1Q^*$ is employed as the reactant. This difference in energy appears to result from steric repulsion between the trifluoroethyl group and the six-membered ring and agrees well with experiment findings (for **4Q\*** from ($R$)-**1Q** in a highly stereo-selective way as shown in Fig. 4b). (iii) As going from $1Q_1$ to $3Q_1$, the sequential cascade reaction is exothermic by 2.0 kcal/mol along with the formation of the more stable neutral ketyl radical

species. To understand the factors that contribute to the greater stability of radical species $3Q_1$ than that of $1Q_1$, we performed the natural bond orbital (NBO) analysis of spin density distribution of the above two species (Fig. 4d). The spin density of $1Q_1$ is exclusively localized on the carbon atom of the substrate, while that of $3Q_1$ is partially distributed on the carbon atom with significant radical character on the $O$ atom and the phenyl group. This reveals that the presence of a OH group as a π-donating moiety can significantly stabilize the resultant radical species $3Q_1$, providing an important driving force that makes the formation of ketyl radical species $3Q_1$ thermodynamically favourable and increases the power of the overall process substantially to generate medium-sized ring systems. DFT calculations on an alternative mechanism involving a Cu species have also been performed[49,50], showing that the reaction of the alkyl radical **1Q** with Cu$^{II}$ species is highly energetically unfavourable, as revealed by a 6.1 kcal higher reaction barrier of the forming Cu intermediate than the radical aryl migration process of alkyl

**Figure 5 | Versatile transformations (I).** (**a**) One-pot Schmidt-Aubé reaction for the construction of medium-bridged lactams. (**b**) Access to enantioenriched medium-bridged tertiary amines.

radical intermediate $1Q_1$ (Supplementary Fig. 3). Therefore, all these experimental and calculated results, together with the observed highly efficient radical chirality transfer during the ring expansion, are in support of our initial proposal as shown in Fig. 1c, in which the selective addition of a variety of *in situ* generated radicals to unactivated alkenes could trigger ring expansion via remote 1,4- or 1,5-aryl migration processes.

**Diverse synthetic applications.** An important aspect of this current methodology is that both medium-sized carbo- and heterocyclic ketones and C-X (X = $N_3$, $CF_3$, $P(O)R_2$, $SO_2Ar$, $C_4F_9$) are efficiently constructed. Consequently, the resultant functionalized compounds can serve as a convenient handle to access other valuable medium-sized and related compounds. Medium-bridged lactams, exhibiting unique reactivity dissimilar to that of traditional amides due to the limited $n_N \rightarrow \pi^\star_{C=O}$ conjugation, have been used as important building blocks in accessing pharmaceutical and natural relevant molecules[39]. Therefore, a number of methodologies have been developed for the synthesis of these useful scaffolds[39]. However, convergent one-step or one-pot methods to prepare such complex skeletons are extremely rare especially with high levels of enantiocontrol, and often require the use of inconvenient polycyclic substrates via tedious multistep synthesis with poor selectivity. To demonstrate one important synthetic utility of this methodology, we reasoned that intramolecular Schmidt-Aubé reaction[51,52] of the resultant azido-substituted medium-sized ketones **3** in the presence of an appropriate Brønsted acid might afford medium-bridged lactam derivatives. Our original screening of reaction conditions proved that the treatment of **3A** with TfOH easily afforded the desired bridged lactam **13** in 85% yield, with complete regioselectivity (Supplementary Table 6). To further simplify the reaction protocol, we investigated the possibility of performing the two distinct reaction sequence in a one-pot fashion. To our delight, after optimization of different reaction parameters (Supplementary Table 7), a simple one-pot procedure was realized: alkenyl alcohols **1** were firstly converted into azido-substituted medium-sized ketones **3** in the presence of CuCN (10 mol%) with EtOAc as solvent. Then, after replacement of the solvent EtOAc with

DCM, **3** were subjected to the intramolecular Schmidt-Aubé reaction with TfOH to afford the medium-bridged lactams **13–16** with distinct backbones in 40–62% yields (Fig. 5a). Moreover, the enantioenriched (*S*)-**13** was easily obtained in 60% yield with >99% ee from (*R*)-**1A** (>99% ee) following the one-pot procedure (Fig. 5b). It is encouraging to note that the present process is a rather general protocol for the one-pot synthesis of useful complex medium-bridged lactams in high efficiency, which is clearly complementary to previous conventional synthetic methods[39]. Most importantly, our procedure is also applicable to the synthesis of a novel type of chiral bridged tertiary amines at the bridgehead position, which have a great potential for applications as effective asymmetric organocatalysts or ligands[53] and could be further converted into other alkaloid analogues of potential medicinal importance[39]. For example, the reduction of the resultant (*S*)-**13** with $LiAlH_4$ efficiently delivered the medium-bridgehead tertiary amine (*S*)-**17** in 79% yield without a decrease in the ee value (Fig. 5b).

The easy access to azido-substituted medium-sized ketones prompts us to exploit the synthetic application of the current methodology to the rapid accumulation of analogues of isopavine alkaloids, which possess important pharmacological activities (Fig. 2). As envisioned, the $PPh_3$-promoted intramolecular Staudinger/aza-Wittig reaction of **3**, followed by subsequent $NaBH_4$ reduction of *in situ* generated imines provided amines **18–20** smoothly in 77–92% yields (Fig. 6a). Finally, the amines were directly converted to isopavine analogues **21–23** in 82–93% yields by reductive amination with formaldehyde in the presence of $NaBH_4$ (Fig. 6a). In addition, the resultant ketone group in the obtained medium-ring product inspires us to extend the ring expansion strategy towards lactams via Beckmann rearrangement. An example was the efficient conversion of $CF_3$-containing nine-membered ketone **4A** to the ten-membered lactams **24** and **25** in 15% and 34% overall yields via Beckmann rearrangement of the corresponding *E*- or *Z*-oximes, respectively (Fig. 6b); such synthetically challenging medium-ring lactams are widespread subunits in biologically active natural products and therapeutic agents[54]. Meanwhile, **3A** successfully underwent the click reaction to give the corresponding triazole **26** in 85% yield (Fig. 6c). Furthermore, manipulation of functional groups at other sites of

**Figure 6 | Versatile transformations (II).** (**a**) Rapid accumulation of a small library of isopavine analogues. (**b**) Preparation of medium-sized lactams via Beckmann rearrangement. (**c,d**) Click chemistry and preparation of sulfone.

the ring proved also feasible. For example, sulfur-tethered ketone **4G** could be selectively oxidized to deliver sulfonyl-substituted nine-membered ketone **27** in 62% yield (Fig. 6d).

**Cheminformatic analysis.** We have used a cheminformatic approach involving principal component analysis to evaluate the diversity of our compound library compared with natural products, drug-like compounds and drugs[18] (see Supplementary Figs 4–10 and Supplementary Tables 8 and 9). The results reveal comparably diverse but distinct chemical spaces occupied by our synthetic molecules and similar benzannulated medium-ring natural products. In addition, compared with the naturally occurring counterpart, our compound library displays less overlaps with the chemical spaces determined by drug-like compounds and drugs. Both of these two features indicate a wide and less-explored chemical space spanned by our compound library, thus demonstrating its great potential for future drug discovery.

**Preliminary biological studies.** The structure resemblance to a variety of biologically active compounds also encourages us to evaluate the biological activity of our products. Our preliminary biological studies revealed that compounds **3I**, **13** and **15** exhibited good cytotoxicities against both H1299 (human non-small cell lung carcinoma, with $IC_{50}$ values of 10.8 and 19.5 μM for **3I** and **15**, respectively) and 293T (derivative of human embryonic kidney transformed by adenovirus, with $IC_{50}$ values of 18.5 and 10.9 μM for **13** and **15**, respectively) cell lines (Supplementary Table 10).

## Discussion

In summary, we have developed a strategically novel, general and powerful approach for diversity-oriented synthesis of skeletally- and functionally diverse benzannulated medium- and macro-ring scaffolds through 1,4(5)-aryl migration/ring expansion triggered by addition of the diverse radicals to alkenes. Furthermore, the newly developed protocol provides a facile and straightforward access to useful medium-bridged amine derivatives, such as the analogues of the azocine nucleus and the isopavine family. This protocol features a diverse product scope including a variety of carbo-, oxygen-, nitrogen- or sulfur-containing 8 to 14-membered cyclic ketones and medium-bridged amines with over 37 distinct skeletally complex scaffolds as well as the identification of some compounds with potent activity in anticancer and proliferation inhibition, wide functional group compatibility, high enantioselectivity (up to 99% ee) by a chirality transfer strategy, and readily available substrates and reagents. The mechanism of this novel strategy was also investigated by control experiments and DFT calculations. Considering the broad utility for diversity-oriented synthesis of various medium rings and bridged rings of this new protocol, we further anticipate that this efficient strategy will motivate the design of other related processes for the more efficient synthesis of complex natural products and other bioactive molecules.

## Methods

**General procedure for azidation reactions.** Under argon, a 25 ml Schlenk tube equipped with a magnetic stir bar were charged with **1** (0.3 mmol, 1.0 equiv), **2a** (104 mg, 0.36 mmol, 1.2 equiv), CuCN (2.7 mg, 0.03 mmol, 0.1 equiv) and EtOAc

(super dry, 3.0 ml). The sealed tube was then stirred at 60 °C for 12 h. After completion (monitored by thin-layer chromatography (TLC)), EtOAc (30 ml) was added. The organic phase was washed with saturated NaHCO₃ solution (2 × 5 ml), dried over anhydrous Na₂SO₄, filtered and concentrated to afford the crude product, which was purified by flash column chromatography to afford the corresponding product **3**.

**General procedure for trifluoromethylation reactions (a).** Under argon, a 25 ml Schlenk tube equipped with a magnetic stir bar were charged with **1** (0.2 mmol, 1.0 equiv), **2b** (126 mg, 0.4 mmol, 2.0 equiv), CuCN (1.8 mg, 0.02 mmol, 0.1 equiv) and EtOAc (2.0 ml). The sealed tube was then stirred at 80 °C for 12 h. After completion (monitored by TLC), EtOAc (30 ml) was added and the reaction mixture was washed with saturated NaHCO₃ (2 × 5 ml) solution. The organic layer was dried over anhydrous Na₂SO₄, filtered and concentrated to afford the crude product, which was purified by flash column chromatography to afford the product **4**.

**General procedure for trifluoromethylation reactions (b).** Under argon, a 25 ml Schlenk tube equipped with a magnetic stir bar was charged with **1** (0.2 mmol, 1.0 equiv), **2b** (126 mg, 0.4 mmol, 2.0 equiv), CuCN (1.8 mg, 0.02 mmol, 0.1 equiv) and dioxane (4.0 ml). The sealed tube was then stirred at 60 °C for 24 h. After completion (monitored by TLC), EtOAc (30 ml) was added and the reaction mixture was washed with saturated NaHCO₃ solution (2 × 5 ml). The organic layer was dried over anhydrous Na₂SO₄, filtered and concentrated to afford the crude product, which was purified by flash column chromatography to afford the product **4**.

**General procedure for phosphonylation reactions.** To a flame-dried Schlenk tube equipped with a magnetic stir bar were added **1** (0.2 mmol, 1.0 equiv), **5a/5b** (0.4 mmol, 2.0 equiv) and AgNO₃ (18 mg, 0.10 mmol, 0.5 equiv). The tube was evacuated and backfilled with argon for three times, and then CH₃CN (2.0 ml) was added. The tube was stirred at 80 °C for 24 h and then H₂O (5 ml) was added. The product was extracted with EtOAc (3 × 10 ml). The combined organic layers were dried over anhydrous Na₂SO₄, filtered and concentrated to afford the crude product, which was purified by flash column chromatography to afford the product **6–8**.
For NMR and HPLC spectra, see Supplementary Figs 11–214.

**Data availability.** The X-ray crystallographic coordinates for structures reported in this article have been deposited at the Cambridge Crystallographic Data Centre (CCDC), under deposition number CCDC 1451652 (**4O**), CCDC 1451653 ((R)-**1A**) and CCDC 1451654 ((R)-**4A**). The data can be obtained free of charge from The Cambridge Crystallographic Data Centre via http://www.ccdc.cam.ac.uk/data_request/cif. Any further relevant data are available from the authors on reasonable request.

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

## Acknowledgements

Financial support from the National Natural Science Foundation of China (Nos. 215722096, 21302088), the National Key Basic Research Program of China (973 Program 2013CB834802), Shenzhen overseas high-level talents innovation plan of technical innovation project (KQCX20150331101823702), Shenzhen special funds for the development of biomedicine, Internet, new energy and new material industries (JCYJ20150430160022517) and South University of Science and Technology of China is greatly appreciated.

## Author contributions

L.L., Z.-L.L. and F.-L.W. performed experiments. Z.G. performed DFT calculations. Y.-F.C. and N.W. helped with characterizing all new compounds. X.-W.D. performed cheminformatic analysis. C.F., J.L. and C.H. performed biological studies. B.T. revised the paper. X.-Y.L. conceived and directed the project and wrote the paper.

## Additional information

**Competing financial interests:** The authors declare no competing financial interests.

**Publisher's note**: 

