## [Peer Review File · Nature Communications]

Reviewers' comments:

Reviewer #1 (Remarks to the Author):

Medium-sized and medium-bridged rings are an important class of scaffolds that feature in a wide range of bioactive molecules. However, such molecules are currently underrepresented in small molecule collections due to the significant synthetic challenges associated with their construction. The development of new methods to generate these scaffolds remains of considerable interest, particularly to those in the synthetic and medicinal chemical field, and those in the pharmaceutical industry.

In this manuscript Lui et al. describe a novel and elegant approach to form a variety of medium-sized and bridged motifs, mediated by a radical-based cascade that features aryl migration followed by ring expansion. The scope of this approach has been illustrated well, generating of a library of 37 distinct compounds. The use of readily synthesised starting materials, in addition to the concise nature of the ring formation presents an attractive synthetic new methodology. Furthermore, the authors have also explored an asymmetric application, enabling the formation of enantioenriched scaffolds.

The mechanistic studies and DFT calculations are convincing, and the supplementary information provided is comprehensive including ¹H, ¹³C, ¹⁹F NMR data, experimental and chiral HPLC analytical data and crystal structures of key library members.

In view of the significant contribution of this research to the field, publication is recommended subject to the following corrections being addressed:

General comments and formatting

The article is currently written in a 'report' style and needs to be formatted correctly to fit the Nature Communications guidelines/style. Multiple paragraphs are excessively long and would benefit from being separated to aid the flow of the article. In addition, the article is extremely long (22 pages and over 6500 words including references) which exceeds the 12-page/ 5000-word limit. In this case it should be shortened considerably e.g. through summarising the substrate scope and experimental results.

The heading 'results and discussion' should be separated to 'results' and 'discussion' as per the requirements for Nature Communications articles.

Additionally, the tense changes throughout the article and this is confusing within some paragraphs.

Content

The authors do not detail the importance of scaffold diversity within the introduction, which if included would highlight the significance of the library generated (i.e. the molecules are diverse through their functional group, appendage, stereochemical and skeletal features).

In this case, although the scaffolds appear diverse, the authors may wish to visually summarise the 'diversity' of the library through chemoinformatic analysis i.e. PMI/PCA plots. The scaffolds which have been generated are highly interesting and valuable for screening collections and the diverse nature of the molecules could be further highlighted.

The biological assessment which is included does not appear to be comprehensive. In addition, this information should appear under a separate sub-heading not added at the end of a non-related paragraph. The authors do not justify why the 'randomly selected compounds' 31a, 13 and 15 were tested and why the particular cell lines were chosen.

Figures

- The compound numbering is confusing to the reader and would benefit from some revision.
- The 'table of contents' should not have R1 labelled as this is never modified. Additionally, the positioning of R1, R2 and R3 is different in the table of contents to that in Table 1 and 2, Figures 3 and 5. These should be consistent.
- Overlapping bonds within Figure 2 need to be corrected.
- Figure 5 is confusing and should be separated into two figures and the text also split accordingly. It is also separated from its figure legend.

Typographical errors

There are numerous typographical errors within the manuscript which need to be addressed. Some examples include:

- The introduction paragraph features numerous errors and is confusing to the reader.
- Page 4 paragraph 1 should read 'With this strategy, We report herein the first practical strategy for selective...' and this whole first sentence should be reworded as it is too long.
- Page 7 - 'alkyenyyl alcohols 1 featuring a six-membered ring' and '1M containing an internal alkene'
- Page 10 - 'switching the benzylic alcohol (10) to an aliphatic substrate (1V).'
- Page 13 - '...we expected that....medium-sized ketones...'
- Page 15 - 'Consequently, the resultant functionalized...'
- Page 17 - '(Fig. 5D); such synthetically challenging...'

- Methods - Phosphonylation reactions, the experimental should read 'The product was extracted with EtOAc (3 x 10 mL)' and 'organic layers'

Reference 35 and 39 quote 'Nature' whereas 2, 3 and 18 this is given as 'Nat.' these should be consistent.

Supplementary information:

- Starting materials 1a-1x appear to be novel, yet no spectra for these molecules is included in the supplementary information. If the molecules are not novel, they should include a reference.
- Typographical error in title of Table S2a 'of reaction'
- Table S2c features bold oxygen and iodine atoms
- General information: should read 'q, quartet'

Reviewer #2 (Remarks to the Author):

This manuscript describes a synthetic methodology for the generation of libraries of compounds with medium to large-ish rings. Some evidence for the involvement of radical species is included, as well as a DFT study.

I will focus my comments on the computational work as that is where my main expertise lies. As reported, this aspect of the study is not particularly interesting or original, with much more extensive computational mechanistic studies reported in the literature, e.g. the work on Cu-catalysed Ullmann coupling by the groups of Houk/Buchwald (10.1021/ja100739h) and later followed up by Fu and co-workers (DOI: 10.1021/ja104264v). Indeed, I appreciate that this was a relatively minor addition to back up the mechanistic postulate, but I was left wondering whether metal involvement had at all been considered, and if not, why not.

In terms of the computational methodology, I was somewhat troubled that the main body of the paper indicated that the M06-L functional was used, while the ESI repeatedly referred to B3LYP-D3, including when the optimised geometries are presented. The reason I started looking at this more closely was that selectivity is explained by a 1 kcal/mol energy difference between potentially competing pathways and I was left wondering how robust this result would be to changes in computational approach. Now, experimentally, the bottom-line obviously is that this works, but the computational work presented did not leave me entirely satisfied that all possibilities had been explored, nor that the balance might not have tipped the other way with some exploration of conformational freedom or a different computational approach. I would not necessarily expect calculations to match up with experimental ee, but a bigger energy difference would make conclusions drawn a little more comfortable. At the very least, I would like to see a more extensive discussion of mechanistic possibilities from the point of view of computation, including perhaps an

estimate of computational uncertainty/noise, and stronger links with the experimental results if a metal involvement can indeed be ruled out from that point of view.

As minor comments, I would have liked to see vibrational frequencies for the TS's included in the ESI, some mention of chemical space maps (e.g. by the groups of Beratan and Raymond) could strengthen the case for the synthetic methodology, and a greater focus on the key conclusions one can derive from substrate scope and optimisations, rather than a description of individual results, would certainly enhance this work for me.

Reviewer #3 (Remarks to the Author):

The authors reported here several interesting transformations via novel radical aryl migrations to construct medium-ring motifs. Besides, excellent chirality transfers were observed starting from enantioenriched tertiary alcohols. Potential applications (biological studies) and mechanism elucidation were conducted in this research. The work is well organized and covers a relatively wide range of examples.

I have a few minor comments that may be taken into consideration before publication:

1. For azidation reaction, the author did provide optimization table in SI, however, more information about side products (3Aa' and 3Aa'') is missing. It would help the readers to understand the reaction better if the ratio could be provided in some entries, especially cases with low yields. And the authors did not mention why they changed to new conditions in the case of 3Ma.
2. For trifluoromethylation reaction, in some instances, the yields were rather poor (30-40%). Is it due to competitive 1,2-oxytrifluoromethylation? It would help to understand mechanism, providing 1,2-oxytrifluoromethylation path was considered in DFT calculation.
3. In SI, there are typos in the descriptions of retention time ((S)-4Ab and (R)-4Ab, S44).
4. Did authors ever try substrate with unactivated alkene disubstituted at the internal position?
5. There is a related strategy reported in Angew. Chem. by Zhu group very recently (10.1002/anie.201605130), please cite in revision edition.

Overall, I think these are very remarkable results and I recommend to publish in Nature Communications subject to the above issues being addressed.

Our Responses to the Comments of the Referees

Referee 1

Comment 1: *Medium-sized and medium-bridged rings are an important class of scaffolds that feature in a wide range of bioactive molecules. However, such molecules are currently underrepresented in small molecule collections due to the significant synthetic challenges associated with their construction. The development of new methods to generate these scaffolds remains of considerable interest, particularly to those in the synthetic and medicinal chemical field, and those in the pharmaceutical industry. In this manuscript Liu et al. describe a novel and elegant approach to form a variety of medium-sized and bridged motifs, mediated by a radical-based cascade that features aryl migration followed by ring expansion. The scope of this approach has been illustrated well, generating of a library of 37 distinct compounds. The use of readily synthesized starting materials, in addition to the concise nature of the ring formation presents an attractive synthetic new methodology. Furthermore, the authors have also explored an asymmetric application, enabling the formation of enantioenriched scaffolds. The mechanistic studies and DFT calculations are convincing, and the supplementary information provided is comprehensive including ¹H, ¹³C, ¹⁹F NMR data, experimental and chiral HPLC analytical data and crystal structures of key library members. **In view of the significant contribution of this research to the field, publication is recommended subject to the following corrections.***

Our response: We very much appreciate these comments of the referee and sincerely thank the referee for recommending publication of this work in *Nat. Commun.*, which is a great acknowledgement of the significance of our work.

Comment 2: *The article is currently written in a 'report' style and needs to be formatted correctly to fit the Nature Communications guidelines/style. Multiple paragraphs are excessively long and would benefit from being separated to aid the flow of the article. In addition, the article is extremely long (22 pages and over 6500 words including references) which exceeds the 12-page/ 5000-word limit. In this case it should be shortened considerably e.g. through summarising the substrate scope and experimental results.*

Our response: We thank the referee for bringing this issue to our attention. Normally, the manuscript for an article is 12 pages and 5000 word **for main text**. According to your suggestions, we have shortened the manuscript appropriately and the special editing did not affect any of the conclusions of the published paper. Since we are writing the article in a 'report' style, we will see whether it will meet the criteria when it is formatted into its template. If not, we will consider shortening the manuscript again to fit the guidelines of *Nature Communications*.

Comment 3: *The heading 'results and discussion' should be separated to 'results' and 'discussion' as per the requirements for Nature Communications articles.*

Our response: We appreciate the referee for pointing out this issue. We have separated 'results' and 'discussion' into two parts in the new manuscript.

Comment 4: *Additionally, the tense changes throughout the article and this is confusing within some paragraphs.*

Our response: We thank the referee for informing us these problems and apologize for these mistakes. Following the referee's valuable suggestion, we have reviewed the manuscript carefully and revised the tense properly. Meanwhile, we highlighted the correction in the revised manuscript.

Comment 5: *The authors do not detail the importance of scaffold diversity within the introduction, which if included would highlight the significance of the library generated (i.e. the molecules are diverse through their functional group, appendage, stereochemical and skeletal features). In this case, although the scaffolds appear diverse, the authors may wish to visually summarise the 'diversity' of the library through chemoinformatic analysis i.e. PMI/PCA plots. The scaffolds which have been generated are highly interesting and valuable for screening collections and the diverse nature of the molecules could be further highlighted.*

Our response: We sincerely thank the referee for this valuable suggestion and have conducted principal component analysis (PCA) to evaluate the diversity of our compound library compared with natural products, drug-like compounds and drugs (see Figs. S4-10 and Tables S6 and S7 in the revised Supplementary Information for details). The results reveal comparably diverse but distinct chemical spaces occupied by our synthetic molecules and similar benzannulated medium-ring natural products. Compared with the naturally occurring counterpart, our compound library displays less overlaps with the chemical spaces determined by drug-like compounds and drugs. Both of these two features indicate a wide and less-explored chemical space spanned by our compound library, thus demonstrating its great potential for future drug discovery. We have added the corresponding results into the revised manuscript and Supplementary Information.

Supplementary Table S6. Structural and physicochemical descriptors used in principal component analysis.

Parameter	Description	Method of Determination
MW	molecular weight	ChemDraw Analysis Window
N	number of nitrogens	ChemDraw Analysis Window
O	number of oxygens	ChemDraw Analysis Window

HBD	number of hydrogen bond donors	http://www.molinspiration.com
HBA	number of hydrogen bond acceptors	http://www.molinspiration.com
RotB	number of rotatable bonds	http://www.molinspiration.com
tPSA	topological polar surface area	http://www.molinspiration.com
ALOGPs	calculated n -octanol/water partition coefficient	http://www.vcclab.org
ALOGpS	calculated aqueous solubility	http://www.vcclab.org
nStereo	number of stereocenters (R + S)	Microsoft Excel
R	number of R stereocenters	ChemDraw Show Stereochemistry
S	number of S stereocenters	ChemDraw Show Stereochemistry
nStMW	$n\text{Stereo} \div \text{MW}$ (stereochemical density)	Microsoft Excel
RSdelta	$R - S$	Microsoft Excel
Rings	number of rings	Manual inspection
RngAr	number of aromatic rings	Manual inspection
RngSys	number of ring systems	Manual inspection
RngLg	number of atoms in largest ring outline	Manual inspection
RRSys	$\text{Rings} \div \text{RngSys}$ (ring complexity)	Microsoft Excel

Supplementary Table S7. Standard deviation and percent contribution for each principal component

	PC 1	PC 2	PC 3	PC 4	PC 5	PC6	PC 7	PC 8	PC9	PC1 0
Standard deviation	2.8 78	1.71 0	1.5 70	1.1 99	1.0 58	0.94 1	0.76 8	0.6 33	0.59 0	0.37 8
Proportion of Variance	0.4 36	0.15 4	0.1 30	0.0 76	0.0 59	0.04 7	0.03 1	0.0 21	0.01 8	0.00 8
Cumulative Proportion	0.4 36	0.59 0	0.7 20	0.7 95	0.8 54	0.90 1	0.93 2	0.9 53	0.97 1	0.97 9

Supplementary Figure S9. Biplots and component loadings for principal component analysis (PCA). The biplots for (a) PC1 vs. PC2, (b) PC3 vs. PC2, and (c) PC1 vs. PC3, and (d) component loadings of the 19 original structural and physicochemical descriptors on the first three principal components indicate the influence of each structural and physicochemical descriptor upon the positioning of compounds in the PCA plots (Figure S9). The four most influential parameters on each principal component are highlighted (yellow).

Supplementary Figure S10. Cheminformatic analysis. Principal component analysis (PCA) of 52 our prepared compounds (labeled as 52-M), 27 benzannulated medium-ring natural products (labeled as 27-B), 47 brand-name small molecule drugs of 2006 (labeled as 47-D), 60 diverse natural products (labeled as 60-N), and 20 commercial available drug-like library compounds in the Molecular Libraries Small Molecule Repository (labeled as 20-C) based on 19 structural and physicochemical parameters. The hypothetical average structure for each series (-AVG) is also shown. (a) PCA plot of PC1 vs. PC2. (b) PCA plot of PC3 vs. PC2. (c) PCA plot of PC1 vs. PC3.

Comment 6: *The biological assessment which is included does not appear to be comprehensive. In addition, this information should appear under a separate sub-heading not added at the end of a non-related paragraph. The authors do not justify why the 'randomly selected compounds' 3Ia, 13 and 15 were tested and why the particular cell lines were chosen.*

Our response: We appreciate the referee for the suggestion on bringing the biological assessment into a separate part. We have put this part separately. In our preliminary biological studies, two cell lines were chosen for the anticancer drug test. Each of these cell lines represents a tumor category posing serious challenges for the health of the world population. 293T is derived from human embryonic kidney cells. The number of patients diagnosed with kidney cancer grows at the fastest pace in recent years. Lung cancer (from which H1299 cell line is derived) is projected as one of the most common cancers in 2016 by NCI. On the other hand, our biological assessment included a series of selected compounds based on their skeletal diversity. Actually, we have screened some compounds including **3A**, **3I**, **13** and **15**. So, we have added all of the screened compounds in Table S8 in the revised Supplementary Information. Undoubtedly, in our further biological study, we are interested in systematically screening all these compounds towards more cell lines in purpose of obtaining promising drug leads. Such studies, when completed, should deserve a separate publication if the result is very promising.

Supplementary Table S8. Anti-cancer study of **3A**, **3I**, **13** and **15**.

Compound	IC ₅₀ (μM)	
	293T	H1299
 9 3A	60.9	23.95
 10 3I	76.6	10.8

 13	18.5	136.7
 15	10.9	19.5

Comment 7: *The compound numbering is confusing to the reader and would benefit from some revision. The 'table of contents' should not have R1 labelled as this is never modified. Additionally, the positioning of R1, R2 and R3 is different in the table of contents to that in Table 1 and 2, Figures 3 and 5. These should be consistent.*

Our response: We sincerely appreciate the referee for bringing the ‘compound numbering’ issue to our attention. We have reorganized the compound numbering through the whole manuscript and supplementary information. For example, we changed the labeling of azido-substituted products to ‘**3A-3M**’ from ‘**3Aa-3Ma**’ and CF₃-substituted products to ‘**4A-4X**’ from ‘**4Ab-4Xb**’.

We also thank the referee for pointing out the mistake concerning substituents on aryl ring and olefin moiety and apologize for it. Accordingly, we have revised it in our revised manuscript to make sure that all are in consistent. For example, R¹ is connected to olefin moiety and R³ is removed in the update version such as the following structure **1**.

Comment 8: *Overlapping bonds within Figure 2 need to be corrected.*

Our response: We thank the referee for informing us this mistake and have revised it in the revised manuscript.

Comment 9: *Figure 5 is confusing and should be separated into two figures and the text also split accordingly. It is also separated from its figure legend.*

Our response: We thank the referee for this valuable suggestion and have separated this figure to Figure 5 and Figure 6 and the corresponding paragraph into two paragraphs.

Comment 10: *There are numerous typographical errors within the manuscript which need to be addressed. Some examples include:- The introduction paragraph features numerous errors and is confusing to the reader.*

Our response: We apologize for our carelessness and give our appreciation to the referee for such kind guidance to us. Accordingly, we have carefully reviewed the whole manuscript and polished our writing again.

Comment 11: *- Page 4 paragraph 1 should read 'With this strategy, We report herein the first practical strategy for selective...' and this whole first sentence should be reworded as it is too long.*

Our response: We thank the referee for this good suggestion and have reorganized this sentence as following: “With this strategy, we report herein the first practical strategy for selective and diversity-oriented synthesis of benzannulated 8 to 11(14)-membered cyclic ketones along with concurrent installation of various functional groups from readily available starting materials. This strategy was realized through concerted remote 1,4- or 1,5-aryl migration/ring expansion sequence triggered by radical azidation, trifluoromethylation, phosphonylation, sulfonylation, or perfluoroalkylation of unactivated alkenes.”

Comment 12: *- Page 7 - 'alkenyl alcohols 1 featuring a six-membered ring' and '1M containing an internal alkene'*

Our response: We thank the referee for suggestion of a proper way to exemplify these sentences and have corrected them in the revised manuscript.

Comment 13: *- Page 10 - 'switching the benzylic alcohol (10) to an aliphatic substrate (1V).'*

Our response: Thanks to the referee, we have revised this description as “the reaction was not significantly affected by switching the benzylic alcohol (**10**) to an aliphatic substrate (**1V**).”

Comment 14:- *Page 13 - '...we expected that....medium-sized ketones...'*

Our response: Thanks to the referee, we have changed the tense in that sentence as “...we expected that the stereochemical information of the tertiary alcohol would be completely transferred to the remote new-formed carbon chiral center in a highly stereoselective way...”

Comment 15: *- Page 15 - 'Consequently, the resultant functionalized...'*

Our response: We appreciate the referee's opinion and changed the sentence there as "Consequently, the resultant functionalized compounds..."

Comment 16: - Page 17 - '(Fig. 5D); such synthetically challenging...'

Our response: We thank the referee for this good suggestion and have replaced "the synthetically challenging..." with "such synthetically challenging..."

Comment 17: - Methods - Phosphonylation reactions, the experimental should read 'The product was extracted with EtOAc (3 x 10 mL)' and 'organic layers'

Our response: We appreciate the referee's suggestion and replaced the original description with "The product was extracted with EtOAc (3 × 10 mL). The combined organic layers were dried..." Meanwhile, we changed the corresponding part in supplementary information.

Comment 18: Reference 35 and 39 quote 'Nature' whereas 2, 3 and 18 this is given as 'Nat.' these should be consistent.

Our response: We thank the referee for pointing out these mistakes and have revised them to "Nat."

Comment 19: Supplementary information: - Starting materials 1a-1x appear to be novel, yet no spectra for these molecules is included in the supplementary information. If the molecules are not novel, they should include a reference.

Our response: We thank the referee for bringing this issue to our mind and have pasted the spectra in our updated supplementary information.

Comment 20: - Typographical error in title of Table S2a 'of reaction'

Our response: We appreciate the referee for pointing out this mistake and have revised it as "reaction" in the revised manuscript.

Comment 21: - Table S2c features bold oxygen and iodine atoms

Our response: We thank the referee for this suggestion and have revised them to normal format as depicted below:

Comment 22: - General information: should read 'q, quartet'

Our response: We appreciate the referee's valuable suggestion and apologize for this mistake and have revised this part in General information.

Referee 2

Comment 1: *This manuscript describes a synthetic methodology for the generation of libraries of compounds with medium to large-ish rings. Some evidence for the involvement of radical species is included, as well as a DFT study.*

Our response: We appreciate the referee's summary of the findings in this work.

Comment 2: *I will focus my comments on the computational work as that is where my main expertise lies. As reported, this aspect of the study is not particularly interesting or original, with much more extensive computational mechanistic studies reported in the literature, e.g. the work on Cu-catalysed Ullmann coupling by the groups of Houk/Buchwald (10.1021/ja100739h) and later followed up by Fu and co-workers (DOI: 10.1021/ja104264v). Indeed, I appreciate that this was a relatively minor addition to back up the mechanistic postulate, but I was left wondering whether metal involvement had at all been considered, and if not, why not.*

Our response: Many thanks for this suggesting consideration of mechanism involving a metal complex. We apologize for not addressing this issue in the original manuscript and this suggestion has been followed by performing the additional DFT calculations on the reaction of the alkyl radical **1Q₁** with Cu^{II} species. In these calculations, the broken symmetry solution of M11 method was used to locate the open-shell-singlet-diradical transition state (**TS-SD**). The theoretical study reported by Li and coworkers indicated that “*when one-electron Cu^I-reductant was available, Togni's reagent will be easily reduced via single-electron transfer (SET) to produce CF₃• free radical and Cu^{II} species, and Then the CF₃ free radical facilely attacks the C=C bond, leading to trifluoromethyl alkyl radical intermediate. These two processes of CF₃ free radical generation and C-CF₃ bond formation are thermodynamically favorable. In addition, trifluoromethyl alkyl radical intermediate binding with the Cu^{II} complex took place on the singlet-diradical state potential energy surface (PES), which has lower reaction barrier than that via the triplet/closed-shell-singlet PES (ACS catalysis, 2015, 5, 2458-2468).*” Based on these mechanistic insights, reaction of the alkyl radical intermediate **1Q₁** with Cu^{II} species was performed exclusively on the singlet-diradical state. As shown in Figure S3 in the revised Supplementary Information, transformation of **1Q₁** to **INT** requires the activation free energy of 20.9 kcal/mol via transition state **TS-SD** with the forming Cu-C bond distance being 2.97 Å. In the transition state **TS-SD**, NBO analysis demonstrates that the spin density was dominantly located on the Cu atom (0.62) and C(-0.92) atom with the calculated spin expectation values ($\langle S^2 \rangle$) of 0.99, indicating that a typical singlet-diradical character of **TS-SD**. Comparison of the reaction barriers between the reaction of the alkyl radical **1Q₁** with Cu^{II} species (20.9 kcal/mol, Figure S3) and radical aryl migration process followed by ring expansion without the Cu^{II} species (14.8 kcal/mol, Figure 4d of our response to comment 4) reveals that the former is energetically unfavorable and thus the mechanism involving metal species of titled reactions could be reasonably

ruled out. The above discussion and **Figure S3** have been included in the revised manuscript and Supplementary Information. On the other hand, please kindly note that such reactions with some substrates (for the construction of products **4N**, **4W**, **4X**) could be realized in the presence of organic base as the catalyst. Therefore, in some cases, the mechanism involving metal species should be ruled out.

On the other hand, please kindly note that Houk/Buchwald (10.1021/ja100739h) and Fu and co-workers (DOI: 10.1021/ja104264v) works have already been cited in the revised manuscript as Ref. 49 and 50.

Supplementary Figure S3. The calculated relative free energies (ΔG_{sol}) in 1,4-dioxane with SMD model at the M11/6-31+G**/SDD/Aug-cc-PVTZ level are given in kcal/mol. The selected bond lengths are in Å.

Comment 3: *In terms of the computational methodology, I was somewhat troubled that the main body of the paper indicated that the M06-L functional was used, while the ESI repeatedly referred to B3LYP-D3, including when the optimised geometries are presented.*

Our response: We apologize for the writing errors in the original manuscript. The B3LYP-D3 functional has been replaced by the functional M11 in revised manuscript

and Supplementary Information as to that the latter was employed in the updated calculations (see also the response to comment 4 of Referee 2).

Comment 4: *“The reason I started looking at this more closely was that selectivity is explained by a 1 kcal/mol energy difference between potentially competing pathways and I was left wondering how robust this result would be to changes in computational approach. Now, experimentally, the bottom-line obviously is that this works, but the computational work presented did not leave me entirely satisfied that all possibilities had been explored, nor that the balance might not have tipped the other way with some exploration of conformational freedom or a different computational approach. I would not necessarily expect calculations to match up with experimental ee, but a bigger energy difference would make conclusions drawn a little more comfortable.*

Our response: Many thanks for bringing our attention to the activation free energy difference between two key transition states responsible for the experimental selectivity and for suggesting comparison of density functional-dependent ee values. This concern has been addressed by performing the additional DFT calculations using several other functionals, including M11, M06L, B3LYP, BP86 and B3P86 functionals together with B3LYP-D3 methods previously used in the original manuscript. The results (see below) have been included in the following Table. Among these functionals, M11 functional gave ee values ranging from 83.6% to 90.4% on basis of the calculated energy differences between transition states TS_{1Q1} and TS_{1Q1'} (1.43~1.77 kcal/mol), which were comparable to that found in the experimental one (ee: 96%). In addition, the B3LYP-D3 functional yields an ee value of 87% in the solvent ($\Delta\Delta G_{\text{sol-cpcm}}=1.58$ kcal/mol). However, the other functions such as M06L, conventional B3LYP, BP86 and B3P86 provide poor and even incorrect prediction for the experimental results. As such, the predicted ee value of the titled reactions involving radical species significantly depends on the functionals employed in the calculations.

Table. The calculated activation free energy difference between two key transition states by different functionals

Functional	^a $\Delta\Delta G_{\text{gas}}$	^b $\Delta\Delta G_{\text{sol}}$	^c $\Delta\Delta G_{\text{sol-cpcm}}$
M11	1.77	1.43	1.56
M06L	0.87	0.83	0.80
B3LYP	-0.03	-0.07	-0.22
B3LYP-D3	1.05	1.02	1.58
BP86	0.62	-0.20	0.31
B3P86	0.49	0.09	0.19

a: the Gibbs free energy difference between TS_{1Q1} and $TS_{1Q1'}$ in gas phase; *b*: solvation Gibbs free energy difference was calculated with full optimization using SMD model; *c*: solvation Gibbs free energy difference was estimated as $G_{\text{solv-cpcm}} = E_{\text{solv}}(\text{CPCM-calculated}) + \Delta G_{\text{corr_gas}}$, where $E_{\text{solv}}(\text{CPCM-calculated})$ refers to the solvation single point energy and $\Delta G_{\text{corr_gas}}$ refers to the thermal correction to the free energy of the solute in the gas phase.

Collectively, M11 functional was the most suitable one for mirroring the experimental enantioselectivity both in gas phase and in solvent. Consequently, we recalculated the potential energy surfaces of two reaction pathways responsible for the stereoselective reactions and replaced the original Figure 4d calculated with B3LYP-D3 functionals by the update Figure 4d shown below. Accordingly, the sections about DFT calculations have been carefully organized in the revised manuscript (see Figure 4d and the paragraph marked in yellow on page 14-15 of the revised manuscript):

Fig. 4d, DFT calculation for mechanistic investigation. The calculated potential energy surfaces for aryl migration and ring expansion processes at M11/BS level of theory in 1,4-dioxane with the full optimization using smd model (BS refers to the used basis sets. For C, H, O, F atoms, the 6-31+G** basis set were used and for Br atom, the Aug-cc-PVTZ basis set was used).

Comment 5: *At the very least, I would like to see a more extensive discussion of mechanistic possibilities from the point of view of computation, including perhaps an estimate of computational uncertainty/noise, and stronger links with the experimental results if a metal involvement can indeed be ruled out from that point of view.*

Our response: According to the calculated results in the response for comment 2, some more mechanistic possibilities have been discussed in the revised manuscript, as described in the following text.

“To probe the origin of the observed stereoselectivity, we further investigated the reaction mechanism computationally using M11 method. The generally assumed alkyl sp^3 -carbon-centered radical species $1Q_1$ was chosen as the starting point to locate two reaction pathways responsible for the stereoselective reactions. The calculated results revealed that this reaction occurs stepwise, involving the formation of the bicyclic rings $2Q_1/2Q_1'$, and the ring expansion to result in the formation of intermediates $3Q_1/3Q_1'$ (Fig. 4d). Some notable points from these calculations are as follows: (i) The addition of sp^3 -carbon-centered radical species to the aryl group (from $1Q_1$ to $2Q_1/2Q_1'$) is exothermic. In addition, the reversed process ($2Q_1/2Q_1'$ to $1Q_1$) has higher reaction barriers than the subsequent ring expansion ($2Q_1/2Q_1'$ to $3Q_1/3Q_1'$). As such, reaction barriers of the aryl migration step could be used to mirror the observed stereoselectivity. (ii) The energy of transition state TS_{1Q_1} , leading to the major (*S*)-product $3Q_1$, is lower than that of transition state $TS_{1Q_1'}$ forming the minor (*R*)-product $3Q_1'$, by 1.4 kcal/mol when (*R*)- $1Q^*$ is employed as the reactant. This difference in energy appears to result from steric repulsion between the trifluoroethyl group and the six-membered ring and agrees well with experiment findings (for $4Q^*$ from (*R*)- $1Q$ in a highly stereoselective way as shown in Fig. 4b). (iii) As going from $1Q_1$ to $3Q_1$, the sequential cascade reaction is exothermic by 2.0 kcal/mol along with the formation of the more stable neutral ketyl radical species. To understand the factors that contribute to the more stability of radical species $3Q_1$ than that of $1Q_1$, we performed the NBO analysis of spin density distribution of the above two species (Fig. 4d). The spin density of $1Q_1$ is exclusively localized on the carbon atom of substrate, while that of $3Q_1$ is partially distributed on the carbon atom with significant radical character on the *O* atom and the phenyl group. This reveals that the presence of a OH group as a π -donating moiety can significantly stabilize the resultant radical species $3Q_1$, providing an important driving force that makes the formation of ketyl radical species $3Q_1$ thermodynamically favorable and increases the power of the overall process substantially to generate medium-sized ring systems. DFT calculations on an alternative mechanism involving a Cu species have also been performed, showing that the reaction of the alkyl radical $1Q$ with Cu^{II} species is highly energetically unfavorable, as revealed by a 6.1 kcal higher reaction barrier of the forming Cu intermediate than the radical aryl migration process of alkyl radical intermediate $1Q_1$ (Fig. S3 in Supplementary Information). Therefore, all these experimental and calculated results, together with the observed highly efficient radical chirality transfer during the ring expansion, are in support of our initial proposal as shown in Fig. 2c, in which the selective addition of a variety of in situ generated radicals to unactivated alkenes could trigger ring expansion via remote 1,4- or 1,5-aryl migration processes.”

Comment 6: *As minor comments, I would have liked to see vibrational frequencies for the TS's included in the ESI.*

Our response: We apologize for not including the imaginary vibrational frequencies of the calculated transition states, which have been added in the revised Supplementary Information.

Comment 7: *some mention of chemical space maps (e.g. by the groups of Beratan and Raymond) could strengthen the case for the synthetic methodology, and a greater focus on the key conclusions one can derive from substrate scope and optimisations, rather than a description of individual results, would certainly enhance this work for me.*

Our response: We sincerely thank the referee for this valuable suggestion and have conducted principal component analysis (PCA) to evaluate the diversity of our compound library compared with natural products, drug-like compounds and drugs (see Figs. S4-10 and Tables S6 and S7 in the revised Supplementary Information for details). The results reveal comparably diverse but distinct chemical spaces occupied by our synthetic molecules and similar benzannulated medium-ring natural products. Compared with the naturally occurring counterpart, our compound library displays less overlaps with the chemical spaces determined by drug-like compounds and drugs. Both of these two features indicate a wide and less-explored chemical space spanned by our compound library, thus demonstrating its great potential for future drug discovery. We have added the corresponding results into the revised manuscript and Supplementary Information.

See also the response to Comment 5 of Referee 1.

Referee 3

Comment 1: *The authors reported here several interesting transformations via novel radical aryl migrations to construct medium-ring motifs. Besides, excellent chirality transfers were observed starting from enantioenriched tertiary alcohols. Potential applications (biological studies) and mechanism elucidation were conducted in this research. The work is well organized and covers a relatively wide range of examples. Overall, I think these are very remarkable results and I recommend to publish in Nature Communications subject to the above issues being addressed.*

Our response: We very much appreciate the comments of the referee and sincerely thank the referee for recommending publication of this work in *Nature Communications*.

Comment 1: *For azidation reaction, the author did provide optimization table in SI, however, more information about side products (3Aa' and 3Aa'') is missing. It would help the readers to understand the reaction better if the ratio could be provided in some entries, especially cases with low yields. And the authors did not mention why they changed to new conditions in the case of 3Ma.*

Our response: We thank the referee for bringing this issue to our attention. In designing this project, we assumed that both 3A' and 3A'' might be generated. During screening the reaction conditions, we almost observed no formation of 3A' and 3A''. The low yield of 3A is due to the decomposition of 1A. However, when we optimized the reaction conditions for synthesis of 4N, we observed a 65% yield of oxytrifluoromethylation product 4N' in the presence of Cu(I) catalyst. When we changed to base as the catalyst, the desired ring expansion product 4N can be detected as a single isomer. These results have been discussed in the revised manuscript.

With regard to preparation of 3M, the reaction is quite messy and the desired product was observed in quite low yield in the presence of CuCN for the 1,2-disubstituted alkene substrate 1M. Our screening of additive revealed that 1,10-phenanthroline (1.5 equiv) could enhance the yield of 3M, so we utilized this reaction condition. These results have been added in the revised manuscript.

It should be noted that in the revised manuscript, we have reorganized the compound numbering according to the comment of Referee 1. For example, we changed the labeling of azido-substituted products to '3A-3M' from '3Aa-3Ma' and CF₃-substituted products to '4A-4X' from '4Ab-4Xb'.

Comment 2: *For trifluoromethylation reaction, in some instances, the yields were rather poor (30-40%). Is it due to competitive 1,2-oxytrifluoromethylation? It would help to understand mechanism, providing 1,2-oxytrifluoromethylation path was considered in DFT calculation.*

Our response: We greatly thank the referee for providing such a question. For trifluoromethylation reaction, we have screened a variety of parameters to inhibit the 1,2-oxytrifluoromethylation reaction. For those whose yield is slightly low, almost no 1,2-oxytrifluoromethylation product was formed and the substrate was decomposed. However, in our expansion of substrate scope, we found that the reaction efficiency depends largely on the structure of substrates. Based on these results, at the end of that paragraph we gave a possible conclusion that *tuning the opening-ring size can have a profound influence on the control of the reaction outcome, probably owing to the presence of different favorable conformations and transition states during the reaction* (page 9 in the revised manuscript). For the DFT calculations, we have

performed the additional DFT calculations on the process of the reaction of the alkyl radical **1Q** with Cu^{II} species, which could give 1,2-oxytrifluoromethylation product from this pathway. As shown in Figure S3 in Supplementary Information, transformation of **1Q₁** to **INT** requires the activation free energy of 20.9 kcal/mol via transition state **TS-SD** with the forming Cu-C bond distance being 2.97 Å. Comparison of the reaction barriers between the reaction of the alkyl radical **1Q** with Cu^{II} species (20.9 kcal/mol, Figure S3) and radical aryl migration process followed by ring expansion without the Cu^{II} species (14.8 kcal/mol, Figure 4d in the revised manuscript) reveals that the latter is energetically favorable and the mechanism involving metal species of titled reactions could be reasonably ruled out with a higher reaction barrier.

See also the response to Comments of Referee 2.

Comment 3: In SI, there are typos in the descriptions of retention time ((*S*)-**4Ab** and (*R*)-**4Ab**, S44).

Our response: We thank the referee for bringing this issue to our mind. We apologize for this mistake and have revised this part in the revised manuscript as follows:

(*R*)-**4A**: 98% ee, HPLC analysis [Daicel Chiralpak AD-H, isopropanol/hexane = 10/90, 1.0 mL/min, λ = 214 nm, t_R (minor) = 6.6 min, t_R (major) = 9.2 min].

Comment 4: Did authors ever try substrate with unactivated alkene disubstituted at the internal position?

Our response: We thank the referee for this valuable question. We have synthesized internal disubstituted alkene **1X**. Unfortunately, the trifluoromethylation reaction provided the abnormal product **4X** in only moderate yield. The probable mechanism was shown below:

Comment 5: There is a related strategy reported in *Angew. Chem.* by Zhu group very recently (10.1002/anie.201605130), please cite in revision edition.

Our response: We thank the referee for providing us this important literature, and have cited it as ref 34 in our proposal part.

REVIEWERS' COMMENTS:

Reviewer #1 (Remarks to the Author):

I am happy to recommend that you accept the paper. I've re-read over the latest draft and supp. info. and it is much improved. They have added in all of my suggestions and I think the PCA analysis supports their results, plus the article reads a lot better.

Reviewer #2 (Remarks to the Author):

I would like to thank the authors for engaging so thoroughly with the comments of all reviewers. I am satisfied that the adjustments made to the calculations and data analysis in particular are appropriate and enhance the experimental results presented. Aside from some minor typographical and grammatical errors, which I trust will be dealt with during the publication process, I am now happy to support publication of this work.

Comments of Reviewer 1:

I am happy to recommend that you accept the paper. I've re-read over the latest draft and supp. info. and it is much improved. They have added in all of my suggestions and I think the PCA analysis supports their results, plus the article reads a lot better.

Our response:

We sincerely thank the reviewer for recommendation of publishing our paper on Nature Communications and appreciate the valuable suggestions from the reviewer.

Comments of Reviewer 2:

I would like to thank the authors for engaging so thoroughly with the comments of all reviewers. I am satisfied that the adjustments made to the calculations and data analysis in particular are appropriate and enhance the experimental results presented. Aside from some minor typographical and grammatical errors, which I trust will be dealt with during the publication process, I am now happy to support publication of this work.

Our response:

We appreciate such positive comments from the reviewer and thank the reviewer's support on publication of our work on Nature Communications. We have read through our manuscript and polished the manuscript well.